# Liquid sculpture and curing of bio-inspired polyelectrolyte aqueous two-phase systems

Chongrui Zhang[1], Xufei Liu[1], Jiang Gong[1] & Qiang Zhao ●[1] ✉

Aqueous two-phase systems (ATPS) provide imperative interfaces and compartments in biology, but the sculpture and conversion of liquid structures to functional solids is challenging. Here, inspired by phase evolution of mussel foot proteins ATPS, we tackle this problem by designing poly(ionic liquids) capable of responsive condensation and phase-dependent curing. When mixed with poly(dimethyl diallyl ammonium chloride), the poly(ionic liquids) formed liquid condensates and ATPS, which were tuned into bicontinuous liquid phases under stirring. Selective, rapid curing of the poly(ionic liquids)-rich phase was facilitated under basic conditions (pH 11), leading to the liquid-to-gel conversion and structure sculpture, i.e., the evolution from ATPS to macroporous sponges featuring bead-and-string networks. This mechanism enabled the selective embedment of carbon nanotubes in the poly(ionic liquids)-rich phase, which showed exceptional stability in harsh conditions (10 wt% NaCl, 80 °C, 3 days) and high (2.5 kg/m$^2$h) solar thermal desalination of concentrated salty water under 1-sun irradiation.

The formation of interfaces and compartments between immiscible liquids plays crucial roles in engineering and lives[1-4], such as the broad use of water ~ oil emulsions[5,6]. In addition to water ~ organics two phase systems, Beijerinck et al. discovered that two aqueous solutions can be immiscible depending on structures and concentrations of polymers and macromolecules dissolving in each phases[7,8]. These systems are termed as aqueous two-phase systems (ATPS), which differ from classic water ~ oil emulsions since NO organic solvents are involved in ATPS. Early examples of ATPS include mixtures between polyethylene glycol (PEG) and dextran aqueous solution[9], and/or PEG/gelatin solutions[10]. Complex coacervates, formed due to electrostatic interactions between oppositely charged polyelectrolytes, are also seen as a sub-class of ATPS, since immiscible aqueous phases are formed as well[11-13]. Research in ATPS have been developed rapidly, allowing for versatile applications encompassing encapsulation[14,15], microfluidics[16], wet adhesives[17,18], and interfacial reactions[19], etc.

Since water is the only solvent in biology, it has been expected that ATPS should play crucial roles in biological systems[20], e.g., all aqueous cellular compartments might possibly be governed by ATPS mechanisms. Such imaginations have not been concretely verified, until recent studies showing that proteins and polynucleotides form ATPS both

in vivo[21] and in vitro[22]. For example, intrinsically disordered proteins form ATPS condensates[23], serving as membrane-less organelles[21] that determine pathology[24] and biology functionalities[25,26]. Compared to ATPS made from synthetic polymers, ATPS existing in biology are defined by stimuli-responsive phase status and cross-phase evolution. That is, both the phase status and microstructures of ATPS are tunable and curable to fit task-specific purposes, an intriguing ability arising from the complicated chemical structures and supramolecular interactions of proteins[27-29]. For example, *mussels* hang themselves to underwater rocks through mussel foot threads and plaques[25,30], which are characterized by high strength, extensibility, and abrasion resistance[31]. Mussel threads are formed through two crucial steps: first, the responsive formation of self-coacervate (i.e., ATPS) of mussel foot proteins (mfps) mixtures in the groove mold, and second, the pH-dependent curing and microstructure replication of mfps ATPS[32,33]. Interestingly, hierarchical pores are found in mussel threads and plaques, which improve wet adhesion strength[34]. While the mfps ATPS are formed through cation ~ π interaction[35], coacervate curing is associated with the oxidative crosslinking of catechol groups[32]. Other than these understandings, little is known about how the ATPS' liquid phases evolve in sea water. In a broader context, most of current research focus

[1]Key Laboratory of Material Chemistry for Energy Conversion and Storage (Ministry of Education), School of Chemistry and Chemical Engineering, Huazhong University of Science and Technology, Wuhan 430074, PR China. ✉e-mail: zhaoq@hust.edu.cn

on liquid-status of ATPS, and it remains elusive to translate their liquid structures to functional solids[36,37]. To these ends, polymers analogous to mfps should be designed to engineer ATPS curing, but it is a grand challenge to resemble protein's functionality with simplified polymers and synthesis[38,39].

Poly(ionic liquids) (PILs) is a sub-class of polyelectrolytes synthesized by the polymerization of ionic liquid monomers[40,41]. Considering the rich variety of ionic liquids, PILs are known for their structure versatility and rich supramolecular interactions at inter-/intra chain levels[42]. Recently we shown that supramolecular interactions among phenyl-containing poly(ionic liquid) copolymers could be tuned to prepare coacervate, which fail to cure as phenyl groups are incapable of efficient crosslinking at ambient conditions[17]. Here we design a task specific PILs which features two unique properties to construct, modulate and sculpture ATPS' liquid structures. The polymer contains nitrile groups which likely form cation ~ π or cation ~ dipole interaction for ATPS, but also undergo efficient cyclization condensation at base conditions. By this design, cross-linkable ATPS and its structure sculpture is rendered possible, leading to the translation of ATPS liquid structures to macroporous hydrogels. Currently,

polymer hydrogels with hierarchical pore channels are being intensively studied for solar thermal evaporation[43,44]. These hydrogels were normally prepared by template methods or repeated freezing drying processes, followed by crosslinking to improve their water stability[45-47]. Through the structure sculpture and curing of ATPS, in this work PIL macroporous hydrogels were rapidly prepared at room temperature, which show structure stability and anti-salt properties. Moreover, carbon nanotubes (CNT) were selectively sequestered in the PIL phase, thus conferring good light absorption to the hydrogel. The resultant polyelectrolyte sponge show high (2.5 kg/m²h) and stable solar thermal desalination performance under 1-sun irradiation.

## Results and discussion
### Preparation and characterizations of bio-inspired ATPS
As shown in Fig. 1a, mussel foot proteins (mfps) contain an abundance of lysine, tryptophan, and dopa[48,49]. Recent studies show that the cation ~ π interactions between lysine (R) and tryptophan (F) amino acids drive the formation of mfps ATPS in seawater[35,50]. Then the catechol moieties (Y) in mfps undergo self-oxidative polymerization which convert ATPS to solid threads and plaques[51]. The formation and

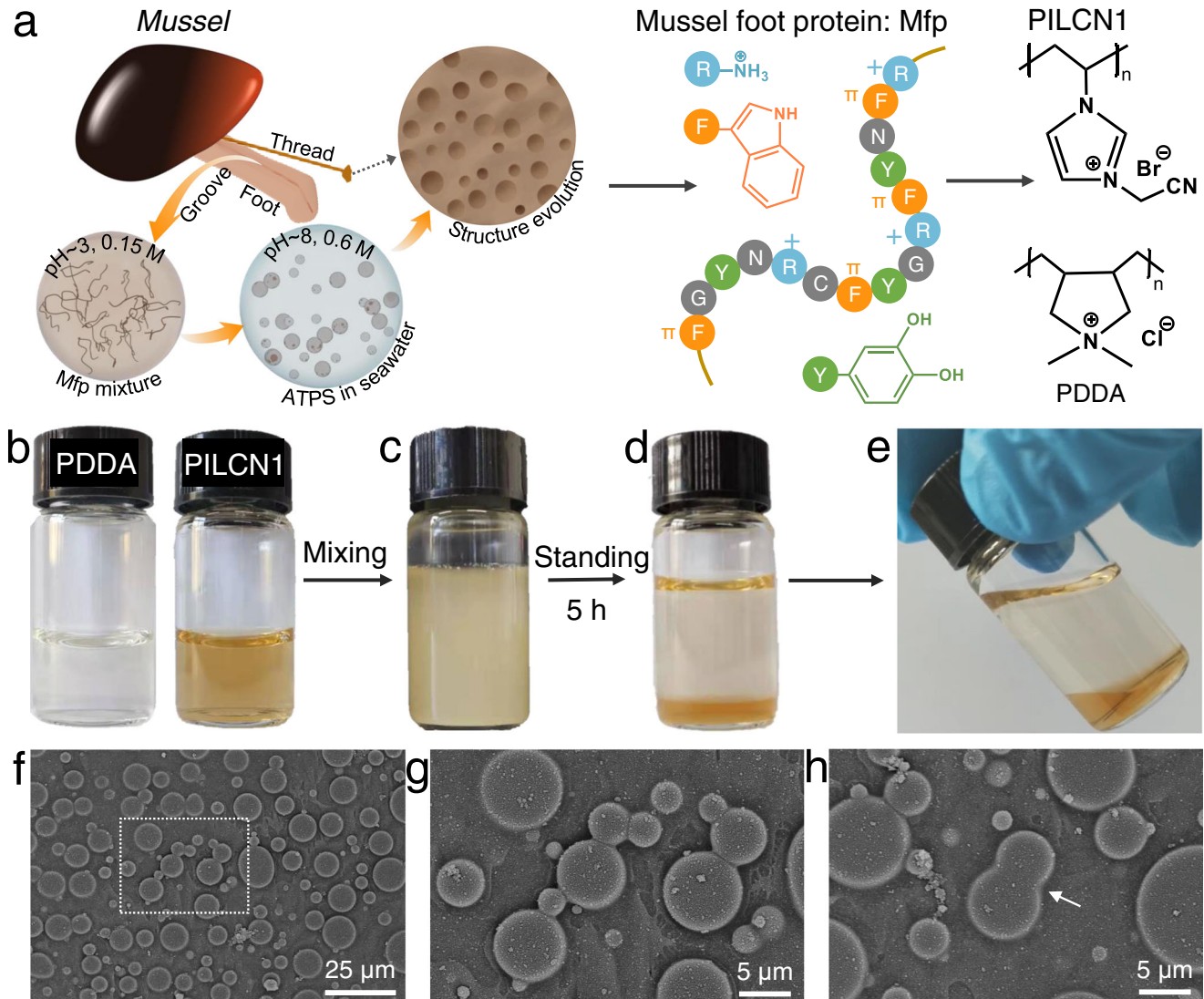

**Fig. 1 | Bio-inspired design and characterizations of ATPS. a** Schematic illustration of mussel foot proteins (middle), the formation and evolution of mfps ATPS (left), and chemical structures of PILCN1 and PDDA (right). **b** Aqueous solution of PDDA (60 mg/mL) and PILCN1 (100 mg/mL). **c, d** ATPS of PILCN1 and PDDA mixture before and after 5 h standing at 20 °C. **e** Optical photographs of the ATPS tilted at 20 °C. **f** Low-magnification Cryo-SEM image of ATPS obtained in **c**. **g** Zoom-in image of **f**. **h** A representative Cryo-SEM image showing the merging of two droplets.

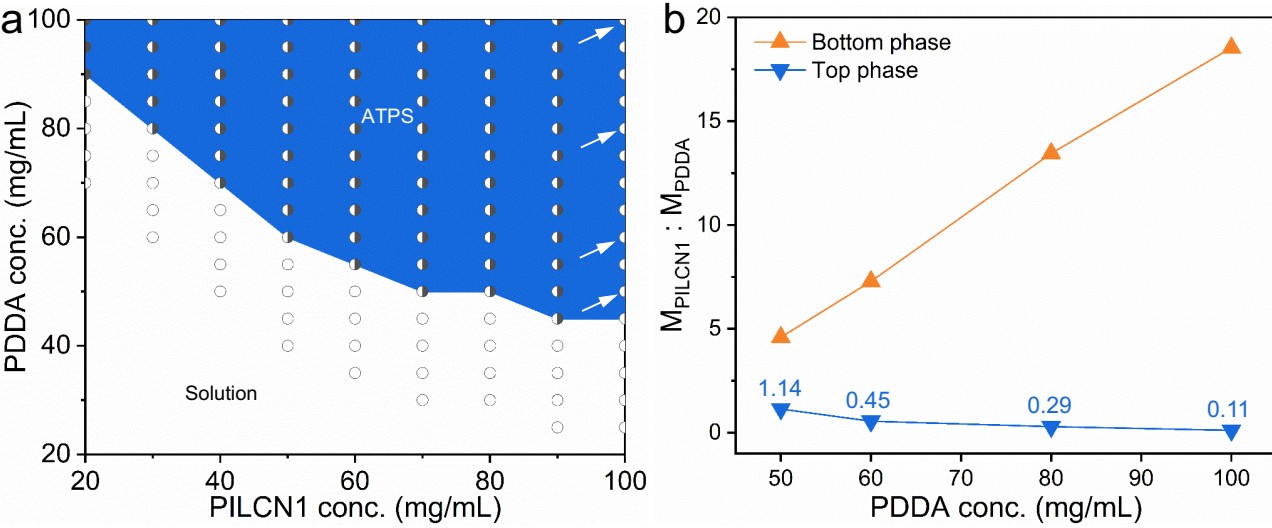

**Fig. 2 | Phase diagram and compositions of the ATPS. a** Phase diagram of the PILCN1-PDDA mixture at 20 °C, Note: PDDA was mixed with PILCN1 at 1:1 volume ratio. **b** Effect of the PDDA concentration in the compositions (mass ratio) of the PILCN1-PDDA ATPS. Note: ATPS samples in **b** correspond to data marked by arrows in **a**.

evolution of mfps ATPS proceed step-by-step at ambient conditions[33]. We confer the stimuli responsive ATPS and curing properties to one single polymer, such as poly(1-cyanomethyl-3-vinylimidazolium bromide) (abbreviation: PILCN1, see Supplementary Figs. 1–4 for monomer and polymer synthesis). The PILCN1 contains π electrons (nitrile) and cations (imidazolium) which would undergo cation - π or cation - dipole interactions, while nitrile groups are activated for cyclization crosslinking when they are attached to cations through methylene spacers[52]. It is the first time that nitriles are exploited for such interactions and polymer condensation. Other cationic polyelectrolytes, e.g., poly(dimethyl diallyl ammonium chloride) (abbreviation: PDDA) was selected as complementary polymers for ATPS.

PILCN1 and PDDA were dissolved in water to prepare solutions (Fig. 1b). A turbid dispersion (Fig. 1c) was obtained when the PILCN1 solution (1 mL, 100 mg/mL) was mixed with PDDA solution (1 mL, 60 mg/mL). After 5 h standing, liquid-liquid phase separation was observed and the bottom phase appears yellowish (Fig. 1d) and fluidic (Fig. 1e, Supplementary Movie 1). Cryogenic scanning electron microscopy (cryo-SEM) characterizations (Fig. 1f) show that the turbid dispersion consists of micron-scale droplets with relatively broad size distributions. These droplets are mainly spherical (Fig. 1g), and some of them are fusing together to form bigger droplets (Fig. 1h, arrow). The fluidity and droplet fusion were also visualized in situ by optical microscopy (Supplementary Fig. 5, Supplementary Movie 2), in agreement with cryo-SEM results. These results confirm the formation of PILCN1-PDDA ATPS.

Phase diagram of the PILCN1-PDDA mixture was studied by mixing equal volume of PILCN1 and PDDA solutions (Fig. 2a). The critical concentration of PILCN1 solution for the solution-to-ATPS transition is inversely correlated to the PDDA concentration, which is consistent with typical phase diagrams reported in literature[53]. Then the PILCN1 concentration was fixed at 100 mg/mL, and the effect of PDDA concentration in compositions of the bottom and top phases of corresponding ATPS were characterized (Supplementary Tables 1, 2). When PDDA concentration of the starting solution increases from 50 mg/mL to 100 mg/mL, the PILCN1: PDDA ratio in the bottom phase increases from 4.59 to 18.53, while that in the top phase decreases from 1.14 to 0.11 (Fig. 2b). That is, there is less PDDA dissolving in the bottom phase (PILCN1-rich) when more PDDA was mixed with PILCN1, and vice versa for PIL dissolving in the top phase (Supplementary Fig. 6).

From the Raman characterization of ATPS (Fig. 3a), PDDA shows peaks at 784 cm⁻¹ and 578 cm⁻¹, which are assigned to C-H vibration of -CH₃ and -CH₂⁻, respectively. Meanwhile, PILCN1 shows characteristic peaks at 2265 cm⁻¹ and 675 cm⁻¹, which are attributed to nitrile groups and the out-of-plane bending of imidazole rings (C-H$_{ring}$), respectively. Interestingly, the 2265 cm⁻¹ and 675 cm⁻¹ peaks decreases and increases, respectively, despite the constant PILCN1 solution concentration. The intensities of both peaks are positively related to the electron density of nitrile groups and imidazole rings[54]. As such, the decreasing and increasing intensity of 2265 cm⁻¹ and 675 cm⁻¹ peaks suggest that the electron density of nitrile groups and imidazole rings were reduced and increased, respectively. Such changes indicate the increasing interaction between imidazolium and nitrile, whereby electrons were shifted from nitriles to imidazolium. To support this explanation, a control PIL (abbreviation: PILeth) containing an ethyl (instead of nitrile) was mixed with PDDA, forming a homogenous solution instead of ATPS (Supplementary Fig. 7). In this case, intensity of the 675 cm⁻¹ peak is stable with increasing the PDDA concentration (Fig. 3b, c), which indicates negligible interactions due to the absence of nitrile groups.

To further verify the supramolecular interaction between nitrile and imidazolium groups, PILCN1-PDDA solution mixture was characterized by 2D ¹H-¹³C heteronuclear single quantum coherence (¹H-¹³C HSQC), which can demonstrate intermolecular correlation signals between different groups. As seen in Fig. 3d, the black arrow indicates the intermolecular correlation peak between the ¹³C moiety of nitriles (115 ppm) and ¹H moieties of imidazole rings (7.5 ppm). This signal confirms a nanometer-scale proximity between the nitriles and imidazole cations[55], which indicates an attractive interaction between nitrile groups and imidazolium.

## Mechanism of ATPS formation

Figure 4a schemes the formation of PILCN1-PDDA ATPS on basis of supramolecular interactions between nitriles and imidazolium (Fig. 3). The interaction was initially counteracted by the charge repulsion between likely-charged PILCN1, thus a homogenous PILCN1-PDDA solution form at lower polymer concentration (Supplementary Fig. 7). When PDDA concentration in starting solutions increases, the ionic strength in the mixture increases, and screens charge repulsion between PILCN1 chains[11,13]. Hence the cation - π interaction become

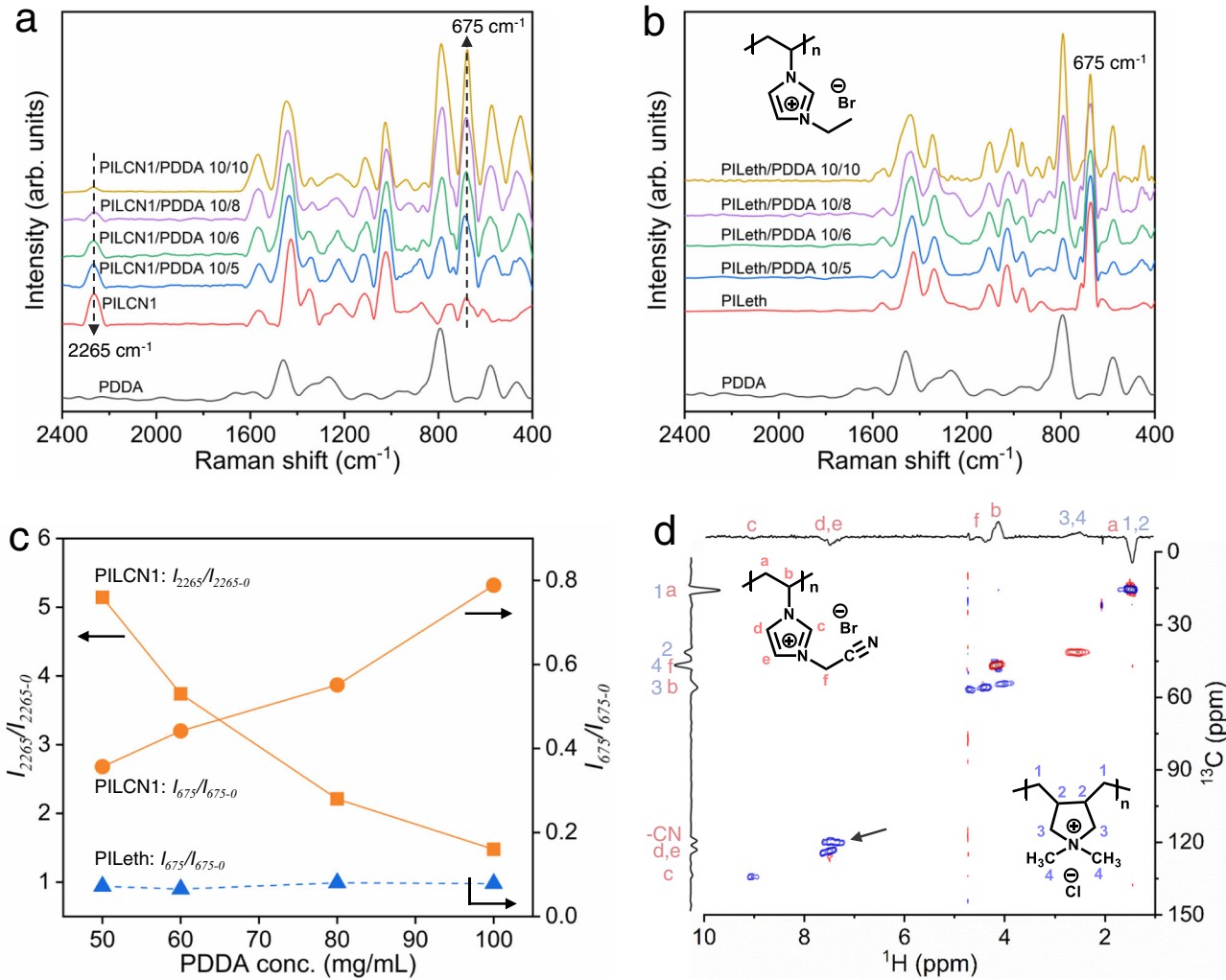

**Fig. 3 | Characterization of interchain interactions.** Raman spectra of **a** mixtures of PILCN1 (100 mg/mL) and PDDA (0, 50–100 mg/mL) and **b** mixtures of PILeth (100 mg/mL) and PDDA (0, 50-100 mg/mL). **c** Effect of PDDA concentration in the Raman peak intensity ratio of PILCN1-PDDA ATPS and PILeth-PDDA mixture solution. Note for peak intensity ratio ($I_x$: $I_{x-0}$: x denotes wavenumbers, and $I_{x-0}$ denotes the intensity of corresponding polymers without adding PDDA). **d** $^1$H–$^{13}$C HSQC characterization of the equal volume mixture of PILCN1 (100 mg/mL) and PDDA (30 mg/mL).

effective to trigger the self-association of PILCN1, leading to the separation of PILCN1-rich (bottom) and PDDA-rich (top) phases.

According to this mechanism, the presence of nitrile groups in PILs are indispensable to drive the cation related interaction and ATPS. Another four PILs were synthesized (Fig. 4b), with nitriles attached to imidazolium rings (PILCN$_1$ - PILCN$_3$) and quaternized ammonium (PILCN$_4$). All these PILCNx (x: 1 ~ 4) form ATPS with three other cationic polyelectrolytes (PDDA, PAE, and PTMAC) at proper concentrations (see Supplementary Figs. 8 and 9 for ATPS characterizations). By contrast, the PILeth polymer, containing NO nitrile groups, forms homogeneous solution with all cationic polyelectrolytes at similar concentrations (Fig. 4b). As further verification of the mechanism, a PILCN1-PDDA solution was turned into ATPS by adding salts into the solution (Supplementary Fig. 10). This is because the added salts screen charge repulsion between PILCN1 chains, which renders the attractive interactions between PILCN1 effective for self-condensation of PILCN1.

**Liquid sculpture of ATPS**

Figure 5a shows that PILCN1 and PDDA solutions (pH 11) were mixed-and-shaken in a round-shape mold at 20 °C, during which the mixture hardened quickly and formed a yellowish hydrogel in 30 s. Compared with both the pristine PILCN1 and PDDA, a new FT-IR peak around

1695 cm$^{-1}$ was seen from lyophilized PILCN1-PDDA hydrogel (Fig. 5b). This peak is assigned to the vibration of -C = N in triazine rings[52], and verifies the occurrence of cyclization reactions of nitriles in PILCN1 during the shaking process. The hydrogel shows good stability in organic solvents (Supplementary Fig. 11), and it was not self-healing due to the covalent crosslinking. Water can be reversibly squeezed out/in from the hydrogel depending on external pressure (Supplementary Fig. 12). Interestingly, both the pristine PILCN1 or PDDA solution alone remain solution state at identical conditions (Supplementary Fig. 13), and the gelation occurred only when PILCN1-PDDA formed ATPS (Supplementary Fig. 14). As shown in Fig. 5a, PILCN1-PDDA ATPS forms up on mixing the two polymers (a-1), and the two fluidic phases were sculptured into bicontinuous phases during the shaking (a-2). Through the formation of ATPS, importantly the concentration of PILCN1 in condense phase is improved to 248 mg/mL, one time higher than the maximum concentration (110 mg/mL) of PILCN1 that are dissolvable alone in water at 20 °C. The higher concentration of PILCN1 is beneficial and imperative for nitrile cyclization, which facilitates the rapid and selective curing of PILCN1-rich phase under base conditions (a-3, Fig. 5b), allowing for the ATPS ~ hydrogel conversion.

Freshly prepared hydrogel (without water washing) was characterized in situ by laser confocal fluorescent microscopy, under which

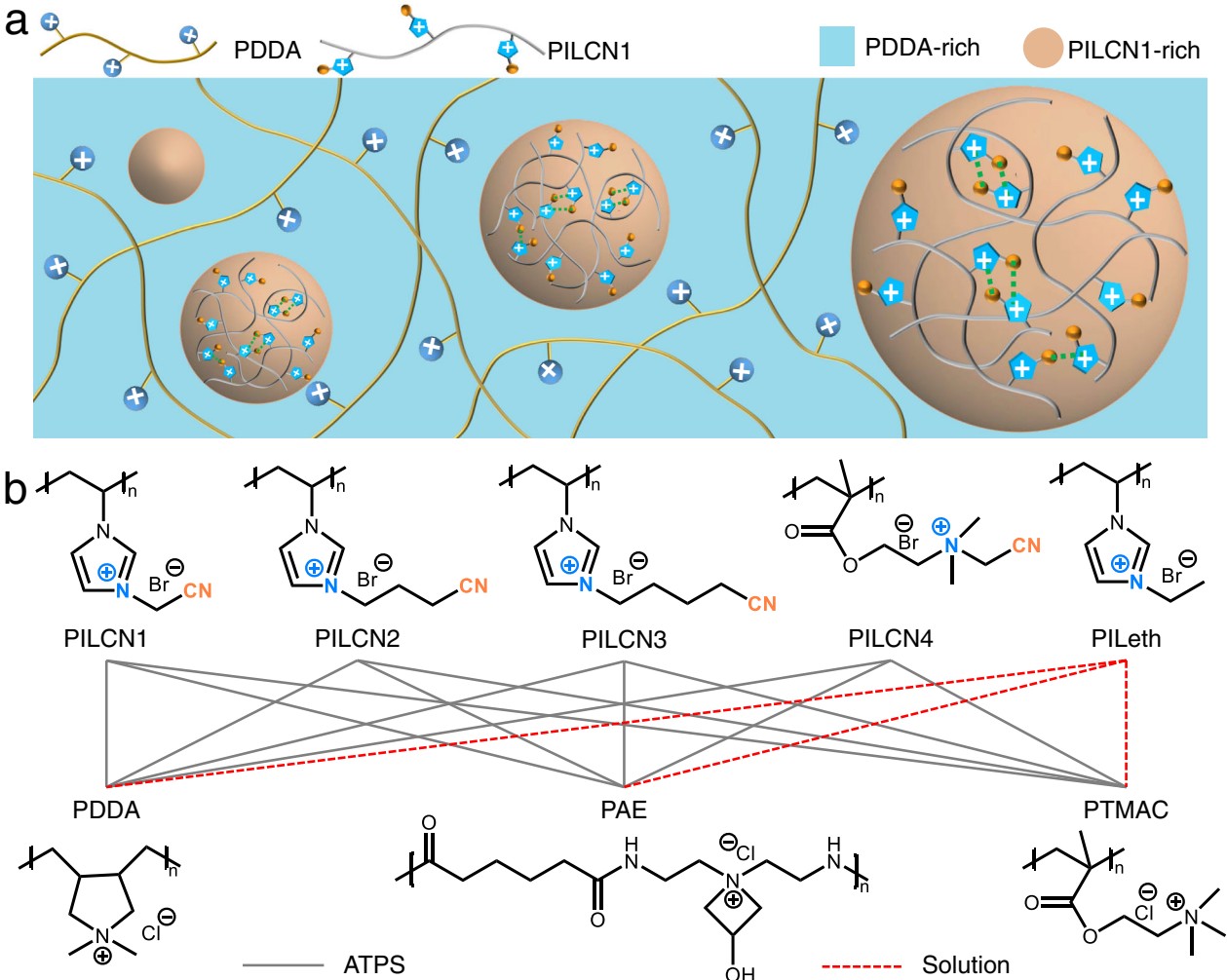

**Fig. 4 | Mechanism of ATPS formation. a** A schematic mechanism of the formation of PILCNx-PDDA ATPS. **b** Chemical structures of control polymers (top row: PILs, bottom row: PDDA, PAE and PTMAC); PILs aqueous solutions (1 mL) were mixed with PDDA, PAE, PTMAC (1 mL), respectively; phase status of the mixture was determined by naked eyes (Supplementary Fig. 8) and optical microscopy (Supplementary Fig. 9). Note: solid and dash lines indicate the formation of ATPS and homogeneous solution, respectively.

PILCN1 is luminescent (Supplementary Fig. 15) while PDDA is not. The orange color in Fig. 5c represents the PILCN1-rich phase, which appear continuous and account for ~13% of the overall area, close to the volume ratio of PILCN1 (Supplementary Table 1). Zoom-in observation (Fig. 5d) shows that the orange phase consisted of round-shaped, interconnected microbeads (dashed line) that are 1–3 microns in size. The lyophilized hydrogel (without wash) shows micro-bead structures that are vaguely visible (arrow, Fig. 5e), as they are covered by PDDA polymers (Fig. 5f). These results verify the formation of PILCN1-PDDA bicontinuous phases during the shaking of ATPS. Alternatively, freshly prepared hydrogel was washed by water, lyophilized and characterized by SEM. From Fig. 5g, the hydrogel shows a sponge-like, macroporous structures, in which the network skeleton consists of interconnected microbeads. Such microstructures were stable after immersing the hydrogel in water (pH 11) for extended time (Supplementary Fig. 16). Figure 5h, i show morphology details of microbeads and how they merge. These beads correspond to the PILCN1-rich phase, which was shaped and rapidly cured while the coexisting PDDA-rich phase was removed during the water wash. As such, the liquid structure of ATPS was sculptured efficiently.

The structure sculpture of PILCN1-PDDA ATPS was further exploited by incorporating task-specific nanofillers. CNTs were dispersed in both the PILCN1 and PDDA solution, and mixed to form ATPS (Fig. 6a).

Similar to results in Fig. 1f, droplets were seen (Fig. 6b, c), corresponding to the PILCN1-rich phase. Interestingly, the PILCN1-rich droplets are black, forming a clear boundary (dash line, Fig. 6c) with PDDA-rich phase. After standing for 10 h, the bottom phase is black, while the top phase is light yellow and transparent, containing very little CNTs (Supplementary Fig. 17). These results show that CNTs are selectively condensed in the PILCN1-rich phase. TEM characterizations show that both the pristine CNT dispersion and PDDA@CNT dispersion consist of clean CNTs with little decorations by polymers (Fig. 6d, e). When it comes to the PILCN1@CNT dispersion, CNTs were covered by more PILCN1 (Fig. 6f), indicating that CNTs have stronger interaction with PILCN1 that is more hydrophobic compared to PDDA[56].

The PILCN1-PDDA ATPS was cured at pH 11 to prepare CNT-containing hydrogels (abbreviation: CNT-hydrogel). Compression strength of the CNT-hydrogel with 0.65 wt% CNT is about 60 kPa and shows exceptional recovery with little hysteresis in 50 cycles of repeated tests (Supplementary Fig. 18). As shown in Fig. 6g, morphology of the CNT-hydrogel is analogues to that of the CNT-free hydrogel (Fig. 5g), i.e., it also features bead (1–4 μm)-thread networks and sponge-like pores (10–30 μm). Such macroporous structures could facilitate the rapid transport of water in the hydrogel (Supplementary Fig. 19). Figure 6h shows the connection of various microbeads, which is due to the droplet fusion arrested by the rapid curing. The tight

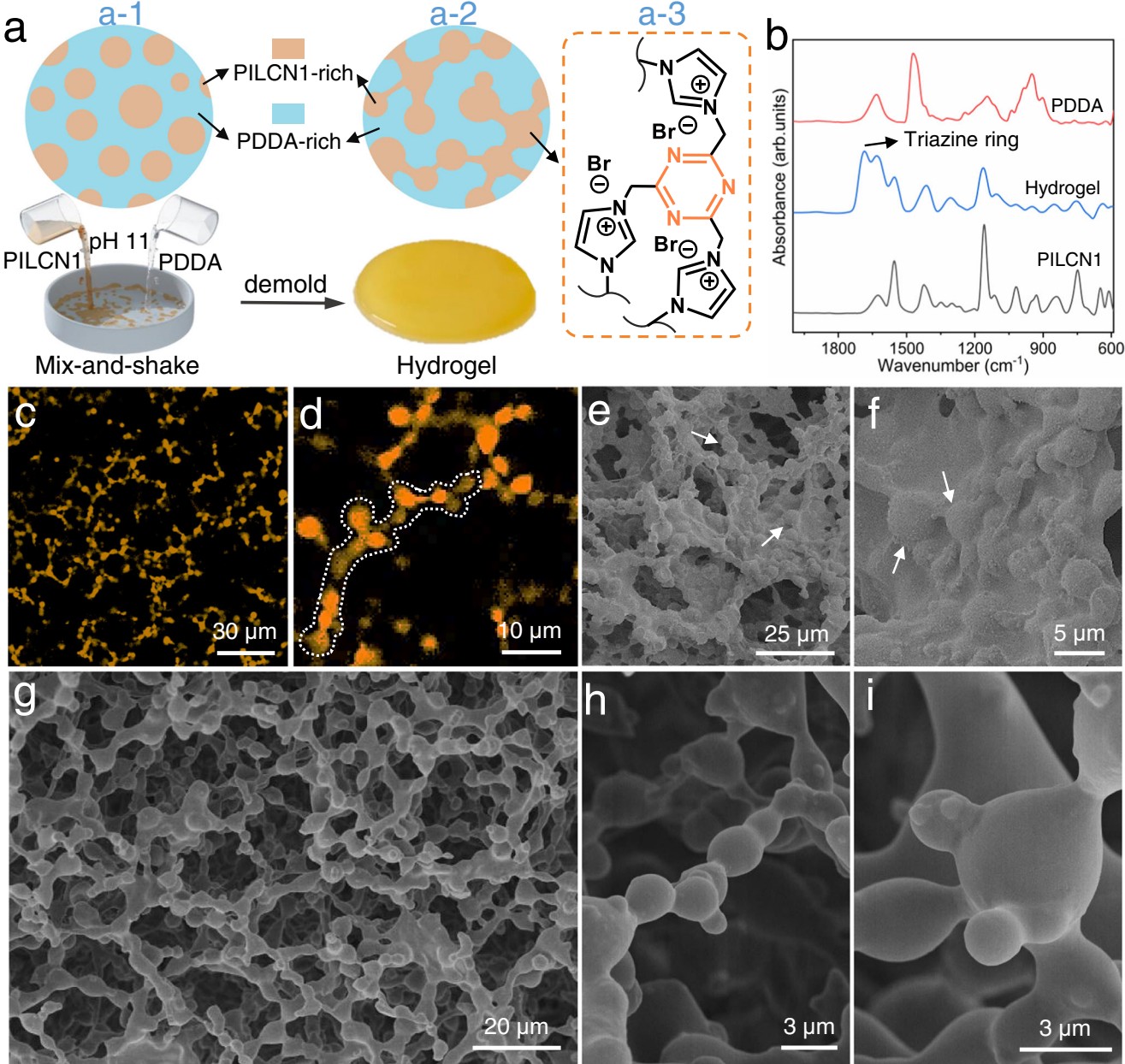

**Fig. 5 | Phase evolution and liquid sculpture of ATPS. a** PILCN1 (1 mL; 100 mg/mL, pH 11) and PDDA solutions (1 mL; 60 mg/mL, pH 11) were mixed-and-shaken for 1 min at 20 °C. **b** ATR-FTIR spectra of PDDA, PILCN1 and lyophilized hydrogel. **c**, **d** Laser confocal fluorescent microscopy images of PILCN1-PDDA hydrogel (405 nm light excitation). **e**, **f** SEM images of as-prepared hydrogel that was lyophilized without washing. **g–i** SEM images of the lyophilized hydrogel that was washed by water for 5 h prior to the lyophilization.

embedment of CNTs in PILCN1-rich phase was clearly verified by zoom-in examinations (Fig. 6i), while NO CNTs were seen in the void space outside of the PILCN1 phase. Moreover, the PILCN1-PDDA-CNT ATPS microbeads were freezing dried to visualize the internal structures of particles. Figure 6j shows that CNTs were seen both at the particle surface and inside it, and a closer observation indicates that more CNTs were dispersed at the surface region of the particle (Fig. 6k). Thus the added CNTs could stabilize PILCN1 droplets via the Pickering effect. This result indicates the stable incorporation of CNTs in the hydrogel, which is beneficial for solar-thermal desalination under harsh conditions.

The proof-of-concept utility of CNT-hydrogel was evaluated by solar thermal evaporation. Effects of CNT content in evaporation rate of CNT-hydrogels were studied, and the CNT content was chosen as 0.65 wt% (Supplementary Fig. 20). At this condition, the CNT-hydrogel

displays an effective light absorption (95%) in the UV-Vis-NIR regions, which is likely due to the fine dispersion of CNTs and the multiply light reflection among surfaces of microbeads (Supplementary Fig. 21). Under 1-sun irradiation, the surface temperature of CNT-hydrogel floating on water rises to 52 °C in 2 min, and then stabilizes around 57 °C (Fig. 7a), 50% higher than that of the pure water (38 °C). The water evaporation rate of CNT-hydrogel is -2.5 kg m$^{-2}$ h$^{-1}$ at 1-sun irradiation, ca. 11 times and 4 times of the pure water evaporation in dark (0.23 kg m$^{-2}$ h$^{-1}$) and under 1-sun irradiation (0.62 kg m$^{-2}$ h$^{-1}$), respectively (Fig. 7b). The water evaporation rate and microstructure of CNT-hydrogel are stable (Supplementary Fig. 22). On basis of these evaporation data, the solar-to-vapor conversion efficiency of CNT-hydrogel is calculated to be ~95%. Figure 7c shows that the evaporation enthalpy of water in CNT-hydrogel is 1.8 kJ g$^{-1}$, ~20% reduced compared to that of the bulk water (2.2 kJ g$^{-1}$). When it comes to real seawater

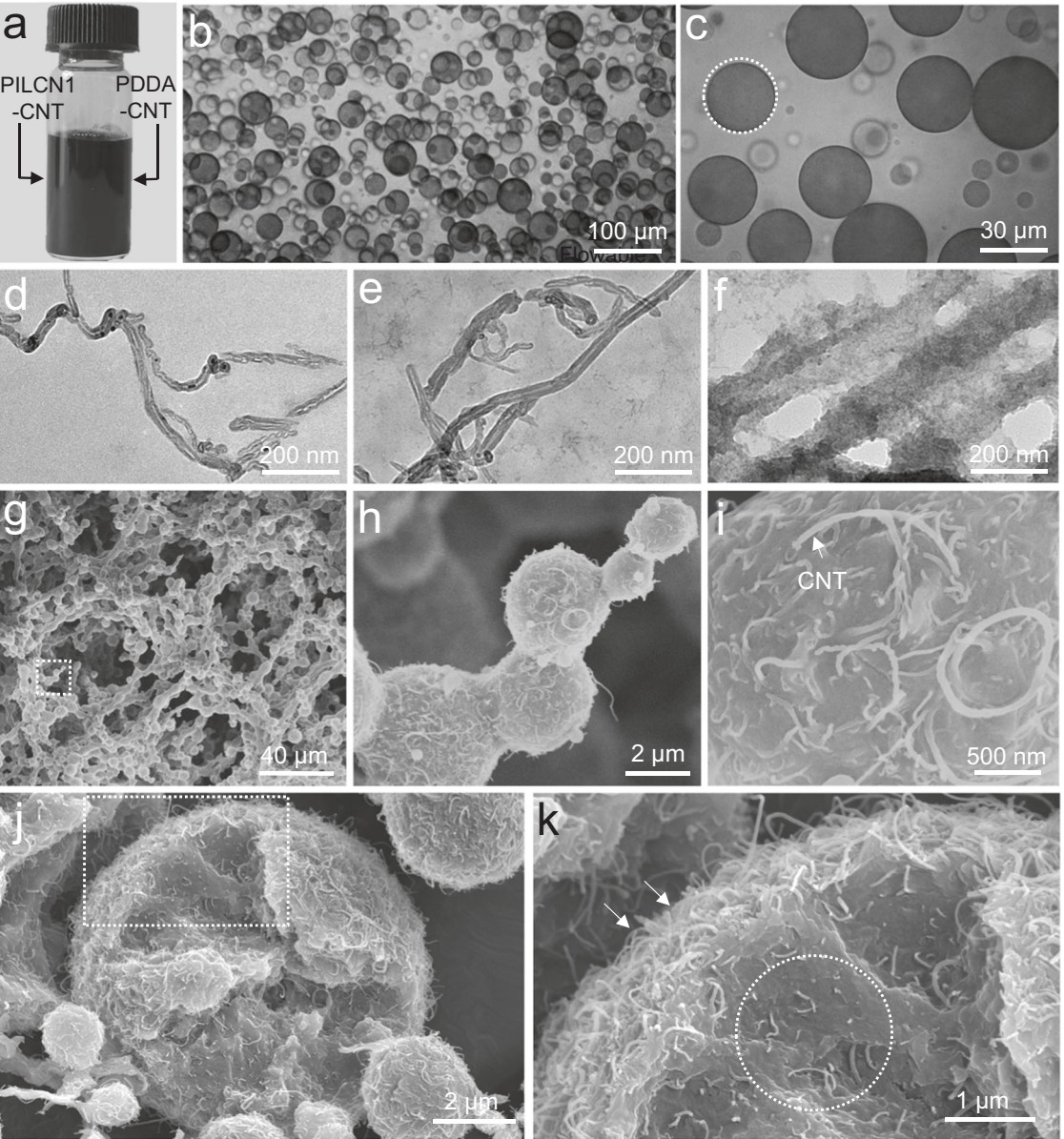

**Fig. 6 | Structure sculpture of CNT-loaded ATPS. a** Photographs and (**b**, **c**) optical microscopy images of mixture of PILCN1-CNT (PILCN1: 100 mg/mL; CNT: 0.25 mg/mL) and PDDA-CNT (PDDA: 60 mg/mL; CNT: 0.25 mg/mL). **d**–**f** TEM images of CNT, PILCN1-CNT and PDDA-CNT dispersions dried on carbon grid, respectively. **g**–**i** SEM images of lyophilized CNT-hydrogel. **j**, **k** SEM images of PILCN1-CNTs particles. Note: the preparation of samples in **j**, **k** was given in section 1.4 of the supplementary information.

desalination, the CNT-hydrogel displays stable evaporation rate (2.5 kg m$^{-2}$ h$^{-1}$) in 10 days running (Fig. 7d, Supplementary Fig. 23). During the seawater evaporation, no salt crystals were seen from the CNT-hydrogel surfaces (Supplementary Fig. 24). Meanwhile, Fig. 7e shows that the concentrations of four major ions (Na$^+$, K$^+$, Mg$^{2+}$ and Ca$^{2+}$) in condensed water are below the drinking water standards defined by the Word Health Organization (1000 mg L$^{-1}$) and U.S. Environmental Protection Agency (500 mg L$^{-1}$)[57].

The salt concentration in water was further increased to 10 wt% (ca. 3 times as concentrated as seawater), and the CNT-hydrogel shows impressive evaporation stability (2.4 kg m$^{-2}$ h$^{-1}$) in 12 tests (Fig. 7f). During the evaporation, only a small amount of salt crystals appeared on hydrogels, which quickly dissolved (Supplementary Fig. 25), indicating that the gel shows good self-cleaning property (Supplementary Fig. 26). The anti-salt and self-cleaning properties of CNT-hydrogels likely arise from the sponge-like pores that accelerate the diffusion and convection of salts, and PILCN1's positive charge that inhibits salt

crystallization[58,59]. In addition, CNT-hydrogel survives the 3-days brine heat treatment (10 wt% NaCl, 80 °C), with little-to-no CNTs washed out of the hydrogel (Supplementary Fig. 27). This exceptional stability arises from the covalently crosslinked PILCN1-rich phase. Considering the high evaporation rate and stable performance in high salinity water, the CNT-hydrogel represents one of the top tier performances among carbon-loaded polymer composites[60], holding considerable potential for solar thermal desalination.

This work engineers the formation and structure sculpture of ATPS between likely charged polyelectrolytes, leading to the efficient preparation of functional hydrogels. When charge repulsion between PILCN1 was screened by external ionic strength, cation-related supramolecular interactions between imidazolium and nitriles drive the self-condensation of PILCN1, leading to the liquid-liquid phase separation of PILCN1 from PDDA. The ATPS evolve into bicontinuous phase structures during shaking, and the concentration of PILCN1 in the PIL-rich phase is two times as high as the maximum concentration of PILs

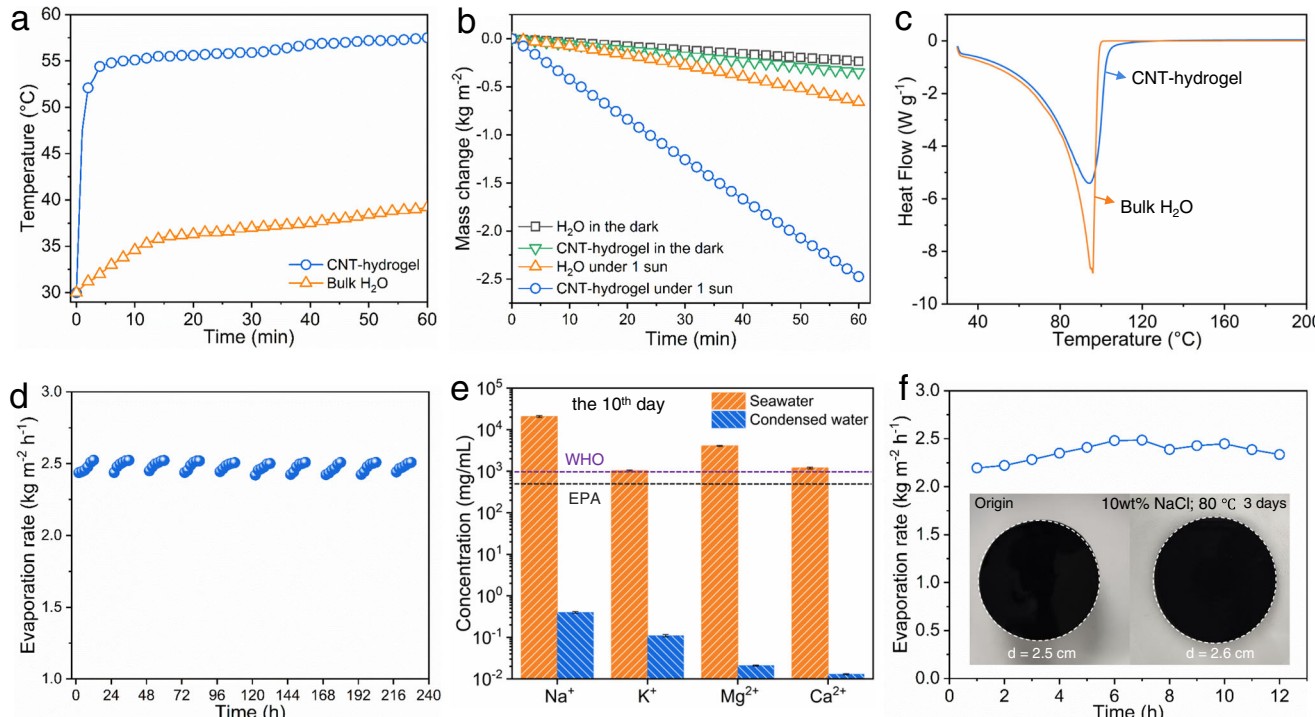

**Fig. 7 | Solar-thermal desalination of CNT-hydrogel. a** Surface temperature of CNT-hydrogel floating on water under 1-sun irradiation. **b** Water evaporation rate of the CNT-hydrogel under 1-sun irradiation. **c** DSC curves of water in the CNT-hydrogel compared to bulk water. **d** Continuous evaporation of seawater using the CNT-hydrogel under 1 sun irradiation. **e** Concentration of ions in pristine seawater and the condensed water obtained from the 10th day evaporation in **d**. **f** Evaporation stability of 10 wt% NaCl aqueous solution using the CNT-hydrogel under 1 sun irradiation (insets: optical photographs of CNT-hydrogel before and after brine heat treatment). Note: error bars in **e** are standard deviations of ion concentration of seawater and condensed water.

dissolvable in water alone. As such, the rapid curing of PILCN1-rich phase was rendered possible through the nitrile condensation at base conditions, resulting in macroporous hydrogels reassembling morphologies of PILCN1 droplets. CNTs were selectively embedded in PILCN-rich phase, and the hydrogel shows high (about 2.4 kg/m²h) solar thermal desalination under 1-sun irradiation. This work shows a paradigm shift in designing responsive ATPS whose liquid structures are modulated and sculptured by pH and salt stimuli.

## Method
### Materials
Polymers (i.e., PILCN1, PILCN2, PILCN3, PILCN4, PILeth, PTMAC) were synthetized by radical polymerization of corresponding monomers, whereby the synthesis details and characterizations were given in Supplementary Figs. 1–4. Poly(dimethyl diallyl ammonium chloride) (PDDA) (Mw = 200,000–350,000 Dalton, 200 mg mL⁻¹ aqueous solution) was obtained from Aldrich. Polyamidoamine-epichlorohydrin (PAE) (125 mg mL⁻¹ aqueous solution) was kindly donated by Golden Huasheng Paper Ltd., Suzhou, China. Multi-wall Carbon Nanotube aqueous dispersion (MWCNT, 10 mg/mL) with a diameter of 30–70 nm and length of 0.3–10 μm was purchased from Nanjing MKNANO Tech Co., Ltd., China. Deionized water (DI water) was produced by a Water Purification System (Easy Heal Force, China).

### Preparation of ATPSs
Aqueous solution of PILCN1 with different concentrations was prepared by dissolving PILCN1 in DI water. PDDA solution with designed concentration was obtained by diluting PDDA solution (200 mg mL⁻¹) with DI water. Equal volume of PILCN1 solution and PDDA solution were mixed at 20 °C, accompanied with occurrence of turbidity in the mixture. Macroscopic liquid-liquid phase separation was observed after 5 h standing of the PILCN1-PDDA mixture. Using similar methods, other PILs were mixed with different cationic polyelectrolytes solution to determine the occurrence of ATPS (Fig. 4).

### Preparation of sponge-like hydrogels
The pH of PDDA solution and PILCN1 solution were adjusted to 11 using 0.1 M NaOH. As schemed in Fig. 5a, equal volume of PILCN1 solution (pH = 11, 100 mg mL⁻¹) and PDDA solution (pH = 11, 60 mg mL⁻¹) were added into the PTFE mold at 20 °C, and the liquid mixture was shaken until gelation occurred. In order to prepare CNT-hydrogel, a designed amount of MWCNTs were first dispersed into PDDA solution and PILCN1 solution, respectively. Then the two solutions were mixed according to protocols described above. As prepared CNT-hydrogels were used for solar thermal steaming tests.

### Solar thermal evaporation measurements
The solar thermal steaming measurement was conducted by a solar light simulator (CEL-S500L) at 30 °C and 50% relative humidity. Typically, the CNT-hydrogels were put on the surface of a 0.5 cm thick polystyrene foam, which was employed as the thermal insulating layer and wrapped by hydrophilic cotton to facilitate water transport. The mass change of water was measured by an electronic balance (JA2003, Soptop). The evaporation rate (kg/m²h) was calculated by: Evaporation rate = $\Delta m / (S*t)$. In this equation, $\Delta m$ is the mass change of water during 1 h evaporation under 1-sun irradiation, $S$ is the area (m²) of the material for evaporation, and $t$ represents the time of solar irradiation. The calculation of evaporation enthalpy was given in Supplementary Information (section 1.6).

### Characterizations
Cryogenic Scanning Electron Microscopy (Cryo-SEM, FEI Quanta 450) and optical microscopy (Olympus IX71, Japan) were conducted to observe the microstructures of ATPS. Raman spectroscopy were

conducted using a Raman spectrometer Nicolet iS50R with 1064 nm laser radiation at 20 °C (the laser power was 0.5 w). 2D-NMR spectrum ($^1$H-$^{13}$C HSQC) was recorded on a Bruker Avance 600 spectrometer. The PILCN-PDDA mixture was thoroughly lyophilized, re-dissolved in deuterated water, and characterized by $^1$H-$^{13}$C HSQC. Up-conversion spectral scanning confocal microscopy (Olympus FV1200) was done with 405 nm light excitation. Field Emission Scanning Electron Microscopy (FESEM) was done with a SU-8010 equipment. The morphologies of CNTs were observed by transmission electron microscopy (TEM, Tecnai G2 20). Attenuated total reflectance Fourier transform infrared spectroscopy was obtained by Nicolet iS5 (Thermo Scientific). Differential scanning calorimetry (DSC) were done with DSC2500 (TA instruments). The inductively coupled plasma emission (ICP-OES, PerkinElmer 8300) was conducted to measure the ion concentration of the condensed water.

## Data availability

Data available on request from the corresponding author.

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

## Acknowledgements

Q.Z. is grateful for the financial support from the National Key R&D Program of China (2022YFB3805103), National Natural Science Foundation of China (22178139), Key R&D Program of Hubei Province (2022BCA079), and Hubei Three Gorges Laboratory (No. SC212001).

## Author contributions

Q.Z. conceived and supervised the project. C.Z. performed the experimental studies with help from X.L. and J.G. All authors wrote and revised the work.

## Competing interests

The authors declare no competing interests.
