## [Peer Review File · Nature Communications]

Liquid Sculpture and Curing of Bio-inspired Polyelectrolyte Aqueous Two-Phase SystemsREVIEWER COMMENTS

Reviewer #1 (Remarks to the Author):

Note from the editor: Reviewer report attached. 
Reviewer #2 (Remarks to the Author):

In this manuscript, the CNT-loaded ATPS composite hydrogel was fabricated for solar vapor generation. There are some issues in the present manuscript. A major revision is needed for improving this work.

1. In the work, the PILCN1-PDDA ATPS was cured at pH 11, and the hydrogel was employed for solar water evaporation. Generally, the pH of water is about 5.5, is hydrogel stable for water evaporation? Whether the hydrogel can be dissolved in water, so can the membrane material be reused in the application?

2. What is the content of CNT in hydrogel? If the amount of CNT changes, how about the solar water evaporation performance of the hydrogel?

3. Since the studied material is a hydrogel and its the ability to enhance the vaporization rate of water, maybe more studies need to be performed. For example, solar-to-vapor conversion efficiency should be provided.

4. Long-term running performance/stability is missing, e.g., continuous one-week running.

5. The mechanical behavior of the CNT-hydrogel is important. Please provide the material's physical properties (compress strength curve) and water absorption rate of material.

6. As a control, the absorption of CNT should be added in Fig. S15.

Reviewer #1

In this manuscript, the authors report a novel bio-inspired strategy to achieve the sculpture and conversion of liquid structures into functional solids. A specifically designed poly(ionic liquid) (PIL) aqueous solution can form an aqueous two-phase system (ATPS) with other cationic polyelectrolytes solutions, such as poly(dimethyl diallyl ammonium chloride) (PDDA), based on “cation \sim π ” interactions between nitrile groups and imidazolium groups. Moreover, nitrile groups are activated for cyclization crosslinking when they are attached to cations through methylene spacers; thus, selective, rapid curing at base conditions led to the evolution from ATPS to macroporous sponges. With the selective embedment of carbon nanotubes in the PIL-rich phase, the hydrogel shows effective solar thermal desalination of concentrated salt. It is an interesting work, and the experiments are well designed.

1. Pls pay attention to Figure 2c. Based on the data in Figure 2a and 2b, it seems that the arrow in Figure 2c points to the wrong y-axis. For the caption of Figure 2c, it should be “PILCN1-PDDA ATPS”, not “PILCN1-PPDA ATPS”.

Response: Thanks a lot. The arrow associated with “PILCN1: I₂₂₆₅/I₂₂₆₅₋₀” line (solid square) was plotted wrongly to the right y axis. This error (Fig. 3c, main text) was corrected. In addition, the typo “PILCN1-PPDA ATPS” was corrected.

2. For the mussel-inspired polymeric materials, self-healing and self-adhesive properties are of great interest. Due to the existence of “cation \sim π ” interaction, how about the self-healing and self-adhesive properties of PILCN1- PDDA hydrogel?

Response: Thanks a lot for this good question. The PILCN1-PDDA hydrogels were covalently cross-linked, thus they were structurally stable, and understandably, NOT self-healing. The PILCN1 does not contain chemical moieties (*e.g.*, catechol groups) enhancing interfacial adhesion. As such, the hydrogel does not show very high self-adhesive properties.

Revision made. Page 11, Lines 2-3. “The hydrogel shows good stability in organic solvents (Supplementary Fig. 9), and it was not self-healing due to the covalent crosslinking.”

3. The mechanical properties of the hydrogel should be characterized.

Response: Thanks a lot. This question was also asked by the reviewer #2. As shown in Figure R1, the compression strength of the CNT hydrogel with 0.65 wt% CNT is about 60 kPa, which shows good reversibility in 50 cycles of tests. The compress strength is comparable to representative macroporous hydrogels in literature [Adv. Funct. Mater. 2020, 30, 2003995; Adv. Funct. Mater. 2019, 29, 1901009], and the good recovery arises from the covalent crosslinking structures (*i.e.*, nitrile cyclization).

Figure R1. Compressive stress-strain curves of the CNT hydrogel with 0.65 wt% CNT.

Revision made: Figure R1 was added as Supplementary Fig. 16.

Page 13, Lines 20-21. “Compression strength of the CNT-hydrogel with 0.65 wt% CNT is about 60 kPa and shows exceptional recovery with little hysteresis in 50 cycles of repeated tests (Supplementary Fig. 16).”

4. A brief background of hydrogels that are used for solar-thermal desalination should be introduced. In comparison to those hydrogels, what are the advantages of the ATPS hydrogel?

Response: Thanks a lot. Please see below the background added in the introduction. In addition, we wish to stress that the primary focus of this work is not the comparison of different hydrogels. Instead we proposed a new type of self-crosslinkable ATPS enabling liquid structure sculpture and materials replication.

Revisions made: Page 3. “Currently, polymer hydrogels with hierarchical pore channels are being intensively studied for solar thermal evaporation^{44,45}. These hydrogels were normally prepared by template methods or repeated freezing drying processes, followed by crosslinking to improve their water stability⁴⁶⁻⁴⁸. Through the structure sculpture and curing of ATPS, in this work PIL macroporous hydrogels were rapidly prepared at room temperature, which show structure stability and anti-salt properties. Moreover, carbon nanotubes could be selectively sequestered in the PIL phase, thus conferring good light absorption to the hydrogel.”

5. In Figure 1d and e, it seems that the PILCN1 can be well mixed with PDDA, which is different with the widely studied PEG-dextran ATPS system. The phase diagram of this new ATPS system is suggested to provided, which is important for understanding the system.

Response: Thanks a lot for this good question. We studied the effect of PDDA and PILCN1 concentration on the phase status of their mixtures, and the phase diagram is shown in Figure R2. The critical PILCN1 solution for solution-to-ATPS transition is inversely correlated to the PDDA concentration, which is consistent with typical phase diagrams reported in literature [Adv. Mater. 2022, 34, 2205649]. Different from the PEG-dextran ATPS, the PILCN1-PDDA ATPS undergoes pH-dependent self-crosslinking, which enables structure sculpture of the ATPS.

Figure R2. Phase diagram of the PILCN1-PDDA mixture at 20 °C.

Revision made: Figure R2 was added as Fig. 2a in the main text.

Page 6, lines 5-9: “Phase diagram of the PILCN1-PDDA mixture was studied by mixing equal volume of PILCN1 and PDDA solutions (Fig. 2a). The critical concentration of PILCN1 solution for the solution-to-ATPS transition is inversely correlated to the PDDA concentration, which is consistent with typical phase diagrams reported in literature⁵⁵.”

6. How about the stability of the hydrogel? Although cyclization condensation at base conditions happens, with time, it is possible that the PILCN1-rich phase would coalesce. Morphology characterization of hydrogel at different time should be given.

Response: Thanks a lot. As shown in Figure R3, sponge-like structures and microbeads of the PILCN1-PDDA hydrogel were maintained when it was immersed in water (pH 11) for different time. In addition, we have shown (Fig. 7f, supplementary Fig. 25) that evaporation performance and microstructures of the CNT-hydrogel were stable after 3-days brine heat treatment (10 wt% NaCl solution, 80 °C).

Figure R3. SEM images of PILCN1-PDDA hydrogel after being soaked in water (pH=11) for (a) 1 h, (b) 6 h and (c) 12 h, respectively.

Revision made: Figure R3 was added as Supplementary Fig. 14.

Page 12, Lines 4-5. “Such microstructures were stable after immersing the hydrogel in water (pH 11) for extended time (Supplementary Fig. 14).”

7. When adding CNTs to the system and producing emulsion droplets, are CNTs dispersed mainly in PILCN1-rich phase or at the interface? The Pickering effects cannot

be ignored.

Response: Thanks a lot for this good question. According to the optical observation (Fig. 6b, 6c), it seems that CNTs were dispersed both in the PILCN1 phase and at the interface. We added SEM observations of the interior of PILCN1 microbeads, which shows that CNTs were distributed both at the interface and inside the particle (Figure R4). Moreover, it seems that more CNTs were distributed at the interface (arrows), though many CNTs were dispersed inside the particle (circle) as well.

Figure R4. SEM micrographs of PILCN1-CNT droplets containing CNT.

Sample preparation: PILCN1-PDDA-CNT dispersion was adjusted to pH 11 without stirring, and the dispersion was kept at pH 11 for 1 h at 20 °C. Then the dispersion was ultra-sonicated (650 W) for 5 mins, purified by centrifugation (8000 rpm) and water washing (3 times), freezing dried, and examined by SEM.

Revision made: Figure R4 was added as Fig. 6j, 6k in the main text.

Page 14, lines 5-9: “Moreover, the PILCN1-PDDA-CNT APTS microbeads were freezing dried to visualize the internal structures of particles. Fig. 6j shows that CNTs were seen both at the particle surface and inside it, and a closer observation indicates that more CNTs were dispersed at the surface region of the particle (Fig. 6k). Thus the added CNTs could stabilize PILCN1 droplets *via* the Pickering effect.”

Reviewer #2 (Remarks to the Author):

In this manuscript, the CNT-loaded APTS composite hydrogel was fabricated for solar vapor generation. There are some issues in the present manuscript. A major revision is needed for improving this work.

1. In the work, the PILCN1-PDDA APTS was cured at pH 11, and the hydrogel was employed for solar water evaporation. Generally, the pH of water is about 5.5, is hydrogel stable for water evaporation? Whether the hydrogel can be dissolved in water, so can the membrane material be reused in the application?

Response: Thanks a lot for this good question. As shown in Figure R5, the solar thermal water evaporation of the CNT-hydrogel is stable around $2.5 \text{ kg m}^{-2} \text{ h}^{-1}$ during 24-h running under 1-sun irradiation. The hydrogel was NOT dissolved in water (Figure R5b, 5c), and maintained stable shape. SEM characterizations show that the hydrogel maintains sponge-like, macroporous structures (Figure R5d-5f). In detail, microbeads structures were maintained (Figure R5e), and CNTs embedment was stable (Figure R5f). Water stability of the CNT-hydrogel arises from the cyclization crosslinking of PILCN1.

Figure R5. (a) Effect of running time on the water evaporation rate of CNT-hydrogel with 0.65 wt% CNT under 1-sun irradiation, (b,c) optical photographs of the CNT-hydrogel evaporator before and after 24 h solar thermal operation in (a), (d-f) SEM micrographs of the CNT-hydrogel evaporator after 24-h solar thermal operation.

Revision made: Figure R5 was added as Supplementary Fig. 20.

Page 16, lines 9-10: “The water evaporation rate and microstructures of CNT-hydrogel are stable (Supplementary Fig. 20).”

2. What is the content of CNT in hydrogel? If the amount of CNT changes, how about the solar water evaporation performance of the hydrogel?

Response: Thanks a lot. The CNT content was 0.65 wt%. As shown in Figure R6a, appearance of the hydrogel becomes more and more dark with increasing the CNT content. Figure R6b shows that the water evaporation rate increases rapidly from 0.8 to 2.5 kg m⁻² h⁻¹ by increasing the CNT amount to 0.65 wt%.

Figure R6. (a) Optical photographs of CNT-hydrogels added with different amount of CNTs. (b) Effect of CNT content in the solar thermal evaporation of the CNT-hydrogel under 1-sun irradiation.

Revisions made: Figure R6 was added as Supplementary Fig. 18.

Page 15, Lines 9. “Effects of CNT content in evaporation rate of CNT-hydrogels were studied, and the CNT content was chosen as 0.65 wt% (Supplementary Fig. 18).”

3. Since the studied material is a hydrogel and its the ability to enhance the vaporization rate of water, maybe more studies need to be performed. For example, solar-to-vapor conversion efficiency should be provided.

Response: Thanks a lot. The solar-to-vapor conversion efficiency was 95%, which was calculated by equations below [ref: Adv. Mater. 2020, 32, 1907061; Sci. Adv. 2019, 5, eaaw5484].

$$E_1 = E_0 * m_0/m_1$$

$$\text{Solar-to-vapor conversion efficiency} = (m_2 - m_1) * E_1 / (3600 * P)$$

Above, E_1 is the evaporation enthalpy of water in CNT-hydrogel, E_0 is the evaporation enthalpy of pure water at 30 °C, which is 2429.8 kJ kg⁻¹, m_0 is the mass of water evaporated in the dark (1 h, 30 °C), m_1 is the mass of water evaporated *via* the CNT-hydrogel in the dark (1 h), m_2 is the mass of water evaporated *via* 1-sun irradiation on CNT-hydrogel (1 h), and P is the irradiation intensity (1 kW m⁻²).

Figure R7. Evaporation rate of H₂O under different conditions.

Revision made: Fig. 7b was replaced by Figure R7.

Page 16, Lines 12-13. “On basis of these evaporation data, the solar-to-vapor conversion efficiency of CNT-hydrogel was calculated to be ~ 95%.”

4. Long-term running performance/stability is missing, *e.g.*, continuous one-week running.

Response: Thanks a lot. We added Figure R8, which shows that the evaporation rate is stable around 2.4 kg m⁻² h⁻¹ during the continuous running. The ion concentrations of condensed water after 10-days running are shown in Figure R9, which is close to data in the main text (original Fig. 6e).

Figure R8. (a) Evaporation rate of seawater *via* CNT-hydrogel under 1 sun irradiation for 10-days. (b) Representative evaporation lines in each day. Please note: lines in (b) correspond to the orange spheres marked in (a).

Figure R9. Concentration of ions in pristine seawater and the condensed water collected from the 10th day CNT-hydrogel evaporation in Figure R8.

Revision made: Figure R8a and Figure R9 were added as Fig. 7d and Fig. 7e, respectively. Figure R8 was added as Supplementary Fig. 21.

Page 16, Lines 15-17 “the CNT-hydrogel displays stable evaporation rate (2.5 kg m⁻² h⁻¹) in 10 days running (Fig. 7d, Supplementary Fig. 21).”

- The mechanical behavior of the CNT-hydrogel is important. Please provide the material's physical properties (compress strength curve) and water absorption rate of material.

Response: Thanks a lot. We added the compress strength curve of the CNT-hydrogel (Figure R10a). The compression strength of the hydrogel is about 60 kPa, which is comparable to representative macroporous hydrogels reported in literature [Adv. Funct. Mater. 2020, 30, 2003995; Adv. Funct. Mater. 2019, 29, 1901009.]. As indicated by Figure R10b, water transported through a freezing dried hydrogel (1 cm height) in about 11 seconds. This result shows the rapid absorption of water in the hydrogel.

Figure R10. (a) Compressive stress-strain curves of the CNT-hydrogel with 0.65wt% CNT. (b) Optical photographs of immersing a freezing-dried PILCN1-PDDA hydrogel (1 cm height) in water containing a trace amount of Rhodamine B dye (for visibility purpose). Please note: To improve the visibility of water absorption, CNTs (black color) were not incorporated in the hydrogel.

Revision made: Figure R10a was added as Supplementary Fig. 16. Figure R10b was added as Supplementary Fig. 17.

Page 13, Lines 20-21. “Compression strength of the CNT-hydrogel is about 60 kPa and shows exceptional recovery with little hysteresis in 50 cycles of repeated tests (Supplementary Fig. 16).”

Page 14, Lines 1-2: “Such macroporous structures could facilitate the rapid transport of water in the hydrogel (Supplementary Fig. 17).”

6. As a control, the absorption of CNT should be added in Fig. S15.

Response: Thanks a lot. The absorption of CNT (black line) was added in Fig. S15.

Figure R11. Absorption of different materials in the UV-Vis-NIR regions.

Revision made: Figure R11 was added in Supplementary Fig. 19.

REVIEWERS' COMMENTS

Reviewer #1 (Remarks to the Author):

The authors have addressed all the comments

Reviewer #2 (Remarks to the Author):

The manuscript has been well revised, which can be accepted by Nature Communication. It is better if the authors can provide the VOC value of the evaporated water.

Reviewer #2 (Remarks to the Author):

The manuscript has been well revised, which can be accepted by Nature Communication. It is better if the authors can provide the VOC value of the evaporated water.

Response: Thanks a lot. The VOC value of the evaporated water is beyond the testing limit of total organic content (TOC, Elementar, Germany) measurement, indicating very low value of VOC. This is because we were using salty water (DI water + NaCl) or natural seawater as the feed for solar thermal desalination, thus it is reasonable that VOC in the evaporated water is very low.